# Sand-Related Factors Influencing Nest Burrowing Potential of the Sand Martins

**DOI:** 10.3390/ani13223463

**Published:** 2023-11-09

**Authors:** Emrah Çelik, Atilla Durmus, Jukka Jokimäki

**Affiliations:** 1Department of Forestry-Hunting and Wildlife Programme, Igdir University, 76000 Igdir, Turkey; 2Ornithology Research and Application Centre (ORNITHOCEN), Igdir University, 76000 Igdir, Turkey; 3Department of Zoology, Department of Biology, Faculty of Science, Van Yuzuncu Yıl University, 65000 Van, Turkey; atilla@yyu.edu.tr; 4Arctic Centre, University of Lapland, 96300 Rovaniemi, Finland; jukka.jokimaki@ulapland.fi

**Keywords:** nesting strategies, soil structure, Collared Sand Martin, Bank Swallow

## Abstract

**Simple Summary:**

Sand Martin populations have declined globally, probably partly because of the decrease in suitable nesting sites. Therefore, it is important to know the factors influencing the nest locations of the Sand Martin. We investigated the physical properties of four Sand Martin colonies, their relationships with the soil characteristics, and the location of the colony (lakeshore sites vs. non-lakeshore sites) in the Lake Van Basin (Turkey). The results indicated that the location of the colony was a critical predictor of the physical properties of nesting holes, including tunnel depth and distance between tunnels. Deeper Sand Martin nest tunnels occur near water sources and soils with higher pH and electrical conductivity values. Nest holes were located closer to each other in lakeshore colonies than in non-lakeshore colonies, and between-hole distances increased with electrical conductivity. The width of the entrance opening increased with soil particle size. The mean particle size in the soil samples collected from nest bottoms was 0.123 mm. Our results indicated that nesting Sand Martins could avoid sites with too compact or loose soils, reducing the possibility of nest collapse. Vertical lakeshore embankments offer good nesting sites for Sand Martins and should be protected.

**Abstract:**

Vertical embankments and mounds serve as suitable habitats for burrowing birds, such as the Sand Martin (*Riparia riparia*). Sand Martins have decreased in many countries during the last two decades, possibly because of the decline in suitable nest sites. Therefore, it is important to understand the factors affecting nest burrowing and nest hole characteristics for the Sand Martin. A smaller entrance hole would be beneficial for regulating the internal environment of the nest, whereas deeper nests are more advantageous against nest predators and parasites. We examined the general structure of Sand Martin colonies and determined if particle size, pH, or electrical conductivity (EC) of the soil and the location of the colony affects the morphology of Sand Martin nest holes. We hypothesized that the climate of near lakeshore and non-lakeshore differs; consequently, we predicted that Sand Martins would construct wider nest tunnel entrances in more humid environments near the lakeshore than further from the lake. We also hypothesized that a lower pH of clay loam soil would result in an increasing level of exchangeable aluminum (Al) and acidity, which in turn would promote soil aggregation. Because soils with a low EC are more stable and less prone to flooding or erosion, we predicted that Sand Martins in such soils would have deeper nesting burrows. A total of four colonies were located in the study area in Turkey. They contained 2510 burrows, of which 91.83% were used for breeding. The mean colony size was 627. We measured the soil and the nest burrow characteristics from the 80 nest bottoms used for breeding by the Sand Martin. The mean pH was 8.8, and the mean EC was 171. Tunnel depth was longer in nests with greater pH and EC and in lakeshore than in non-lakeshore colonies. The distance between nest holes increased with the EC, and nests were located nearer to each other in the lakeshore colonies than in non-lakeshore colonies. The width of the entrance opening increased with soil particle size and was wider in nests located at the lakeshore areas. Our results indicated that Sand Martins will avoid sites with too compact or loose soils for nesting, probably to avoid nest collapses. Vertical lakeshore embankments offer good nesting sites for Sand Martins and should, therefore, be protected. Because soil particle size, pH, EC, and distance from the lakeshore influenced the nest hole characteristics of the Sand Martin, conservation and management efforts should take these variables into account when maintaining or establishing suitable soil conditions for the Sand Martin.

## 1. Introduction

Numerous factors can influence nest selection, including the availability of food and water [1], the presence of predators [2], the quality of the nesting material [3], and other environmental conditions [4,5,6,7]. Therefore, understanding the environmental factors that affect bird nesting habits is important [8]. Some land bird species, like the Sand Martin (*Riparia riparia* [2], Bee-eaters (*Merops apiaster*, and *Merops philippinus*) [9], Common Kingfisher (*Alcedo atthis*) [10], European Roller (*Coracias garrulus*) [11], and Burrowing Owl (*Athene cunicularia)* [12], nest in burrows. In some cases, burrowing birds dig their own burrows, whereas others use burrows abandoned by other animals [12,13,14,15,16]. Studying nest site occupancy and nest characteristics can provide valuable insights into the breeding ecology and population dynamics of burrow-nesting species.

In general, hole- and burrow-nesting species experience lower predation rates than open-cup nesting species [17]. The design and construction of nests significantly affect breeding success [4]. For example, factors such as nest hole depth, spacing, entrance dimensions, and colony formation affect the nesting success of Sand Martins [3,18,19]. Additionally, burrow characteristics (morphology) and sand type influence predation [20,21,22] and nest parasitism in burrow-nesting bird species [23]. However, the relationship between tunnel depth and disturbance may vary among regions and habitats [24].

Resistance to soil erosion is crucial for nest site selection for all three Central European burrowing bird species: Sand Martins, Common Kingfishers, and European Bee-eaters [25]. The depth of the nest tunnels can interfere with parasite detection because the deeper the tunnels, the less sunlight is available [26]. Therefore, it is essential to understand the factors that influence nest hole morphology, especially tunnel depth, of burrow-nesting bird species [27]. Earlier studies have indicated that soil particle sizes can influence nest site characteristics and colony site selection of the Sand Martin [1,3,28,29,30,31,32,33,34,35]. For instance, fine-grained soils, which have a high proportion of small particles, may be easier to dig through than coarser-grained soils [28,29]. Critically, the size of the soil particles can also affect the stability and durability of the burrow over time [30]. However, in addition to the soil particle size, other factors, including the distance to the lakeshore, can significantly affect the behavior of birds, including nesting and feeding habits and response to predation risk. Also, it is possible that soil electrical conductivity (EC) could affect the depth of nest holes in burrowing birds, although this would likely depend on the specific characteristics of the soil and the preferences of the bird species [35].

Soil EC is related to the soil’s water content, mineral composition, and organic matter content [36,37]. Soils with a high EC may be more prone to flooding or erosion [38], which could make them less stable for the burrows of birds. In such cases, burrowing birds may choose to build their nest holes at a shallower depth to avoid these potential risks [39]. A low EC may, therefore, allow burrowing birds to dig deeper nesting holes without negative effects caused by the stability of the soil [24,40].

The difference in nest tunnel depth among bird colonies may also be influenced by soil EC. The impact of soil EC on nest hole depth in burrowing birds is likely dependent on soil characteristics and species preferences. Soil EC is associated with factors such as water content, mineral composition, and organic matter content. Soils with high EC are more susceptible to flooding or erosion, making them less stable for burrowing birds to dig into. Earlier studies have indicated that soils with low EC are more stable and less prone to flooding or erosion. Consequently, we predicted that burrowing Sand Martins in such soils can dig deeper nesting holes without decreased soil stability.

We studied Sand Martin nesting colonies in the Lake Van area in Turkey and related the morphological characteristics of the nesting burrows to soil characteristics (particle size, pH, and EC). The Sand Martin is a small passerine bird species weighing between 11 and 20 g and widely distributed across all continents except Antarctica. Sand Martins are colonial burrowing nesters that build their nests in large groups, often in close proximity to one another [41,42]. The size of a colony can vary greatly, and the largest colonies can contain thousands of nests [28]. Sand Martins are social birds [41], and they benefit from nesting in colonies because it provides protection from predators, which allows them to share information about food sources [1]. Sand Martins also engage in cooperative breeding, where multiple adults help to raise the young in a single nest. This can serve to increase the survival rate of young [20,43]. Sand Martins are known for building their nests in natural vertical banks of rivers, streams, lakes, artificial sand and gravel quarries, and sandbanks by roads [44,45,46]. They excavate their nests in the soft soil of the banks using their bills and feet, creating a tunnel that leads to a small terminal chamber where they lay their eggs [15]. Current studies indicate that Sand Martin populations have declined over the last decades, e.g., in Finland [47] and North America [48,49]. Breeding sites of the Sand Martin are generally ephemeral and can often threatened by human activities, like flood control and erosion control on rivers [48]. Sand Martins typically arrive at their breeding sites in Europe and Asia in April or May, forming nesting colonies in various locations, including soil embankments, sand quarries, and road excavations beside water bodies [50]. Turkey hosts a significant Sand Martin population (100,000–250,000 breeding pairs), making up approximately 5% of the European population, yet it is facing a declining trend [50]. Breeding colonies in Turkey are mainly found in central Anatolia, northern regions, and the inner parts of the country [51]. For our study, all available Sand Martin colonies in the Lake Van Basin (Turkey) were included. Sand Martins return to Lake Van breeding grounds during late April or early May and leave the basin in September [52].

Particle size, pH, and EC can influence soil structure [53,54], consequently influencing the digging properties of nest burrows [54]. We hypothesized that soil pH could potentially impact tunnel depth and penetration resistance. The lower pH of clay loam soil might result in an increased level of exchangeable aluminum (Al) and acidity, which in turn could promote soil aggregation. This enhanced soil aggregation may hinder the Sand Martins from excavating deep nesting burrows. Additionally, we hypothesized that soil pH, EC, and soil particle size can influence the morphometric characteristics of Sand Martin nests. Regarding distance to water, we assumed that the microclimate would influence the morphology of nest holes. We predicted that Sand Martins would construct wider nest tunnel entrances in warmer and more humid environments near the lake than further from the lakeshore because of enhanced ventilation and the reduced risk of overheating. Moreover, colonies near the lakeshore are more susceptible to the effects of surface evaporation from the lake. The resulting water vapor from evaporation can create a certain level of moisture within the colonies, potentially influencing the alkalinity of the soil. Therefore, we predicted that the non-lakeshore colonies would be less affected by the evaporation process.

## 2. Materials and Methods

### 2.1. Study Area

Fieldwork was conducted from 20 April to 23 September 2021 in Van Province, Turkey (Figure 1). Data were collected from four breeding colonies: Çitören Village (CV-1; 38°35′9.78″ N, 43°13′48.07″ E, CV-2; 38°35′8.85″ N, 43°13′49.19″ E), Dilkaya Mound School (DMS; 38°21′33.71″ N, 43°8′27.63″ E), and the Dilkaya Mound (known as Dilkaya “Höyüğü” in Turkish) Coast (DMC; 38°20′37.50″ N, 43°8′6.50″ E).

The climate of Van Province is characterized by cold winters and hot, dry summers. The soil of the study areas is alkaline, which makes it difficult for plants to grow. According to the European Nature Information System classification [55], the study colonies were located on arable lands and inland surface water area habitats (Figure 1).

The non-lakeshore CV-1 and CV-2 colonies were formed because of the construction of a new road in 2013. However, the DMS and DMC colonies are natural embankments at Lake Van. Two of these colonies, CV-1 and CV-2, are in close proximity to each other, with only a 10 m paved road separating them. These colonies are situated far (1 km) from the lakeshore. CV-1 faces north, and CV-2 faces south. DMC and DMS colonies are located on the lakeshore, approximately 1800 m from each other. The DMS and DMC colonies both faced south.

### 2.2. Measurements of Nesting Colonies and Nest Morphology

We systematically searched Sand Martin nesting colonies in the Lake Van study area (5.55 km^2^; Figure 1 and Figure 2) from 2016 to 2021. Surveys have taken place mainly in the coastal areas and basins of Lake Van. During our surveys, we examined areas along riverbanks, lakeshores, rocky areas, and village roads (Figure 1). However, we did not discover any colonies in rocky or shallow lakeshore areas. Colony surveys were conducted on foot along the edges of newly constructed roads and along riverbanks and the lake shoreline. Boundaries of the study area were defined by considering the presence of river canals, village roads, and lake shoreline (Figure 1).

More detailed field studies than searching colonies were conducted during daytime and evening hours between 20 April 20 and 23 September 2021, following the arrival of Sand Martins in the region. Colony size was estimated by a photo of each colony and counting the number of holes on the photo (Figure 2). The number of occupied nests (i.e., nests that were used for breeding during the study period) was determined using two methods. Firstly, we used the method described by Gilbert et al. (1988) [56]. In this method, holes that had traces of birds having passed through them on the entry were considered occupied, and the remainder as unoccupied. Holes with grass or spider webs at the entrances were also categorized as unoccupied holes [57]. When conducting this search, two researchers made nest observations from different ends of the embankment (and met in the middle of the embankment) to collect information about the nest tunnels. Thus, nest holes were checked one at a time, and nest holes were grouped as occupied or empty. In some cases, nestlings were observed at the entrance, and the use of the hole for nesting was easy to determine. Secondly, nestling sounds were used to determine if the hole was used for breeding. During the colony searches, it was determined that it was easy to reach the nests and check their interiors. This ease of access was advantageous for monitoring nest occupancy and collecting soil samples from the nests. However, the proximity of individual nests made it difficult to determine from which hole the sounds came. In these cases, we monitored the colony and observed the holes the parents entered. During the fieldwork, we found that some nesting holes were excavated superficially, and such incomplete tunnels were also classified as unoccupied. Finally, we used the results of both these methods to select occupied holes from which we later collected our soil samples and measured the nest hole characteristics.

The distance of the colonies to the nearest water source, human settlements, and highway was measured using maps. The distance between nesting holes within each colony was measured by a meter ruler. Morphological features of nesting holes were measured using a meter ruler and ladder. The ladder allowed for the assessment of nesting holes located in the upper parts of the soil embankment. The following nest characteristics were measured in the field: the nesting tunnel depth, height of the entrance opening, width of the entrance opening, and length and height of the embankment.

### 2.3. Soil Sampling and Analyses

To investigate the nesting environment of Sand Martins, soil samples were collected from the four colonies (two non-lakeshore colonies, CV-1 and CV-2, and two lakeshore colonies, DMS and DMC; Figure 1 and Figure 2) with 20 randomly selected nesting holes used as breeding (occupied burrows) burrows in each colony. The spacing between sampled holes depended on the length of each colony’s embankment: approximately 1 m for CV-1, 0.60 m for CV-2, 0.40 m for DMS, and 0.75 m for DMC. As the soil barrier lengths of the colonies were not the same, the sampling intervals differed according to barrier length. Sampling intervals that considered the lengths contributed to a homogeneous sampling for each colony. Thus, in all colonies, the nest holes were sampled at regular intervals between the point where the nest holes first began and the point where they ended. We initially divided each colony into sections based on embankment length for uniform representation. In these sections, we established a reference point using a random number generator to begin our selection. Starting from this reference point, we systematically selected nesting holes at specified intervals, ensuring an even distribution along the embankment. In addition, we used the following method to determine sample points based on the length of the bank: For example, if the bank of the DMS colony was 15 m long, 20 nest samples at 0.75 m intervals were taken. We applied this procedure along a single linear axis and by sampling from the base, center, and immediately above the mound. We used this repetitive approach consistently across all colonies. We did not sample nest holes exclusively in a linear direction.

The sampling was conducted during the active nesting period for each colony. The soil was sampled just below the nesting holes, not exceeding 2 cm depth, and at an average depth of 1 cm from the surface. Each soil sample weighed 150 g [29]. Soil samples were not collected from within the tunnel holes as these tunnels already housed nests and were not disturbed. Sampling was also not performed from the accumulated soil beneath the soil barriers because this material was sourced from multiple nesting holes and did not exhibit a homogeneous distribution. Therefore, it was deemed more appropriate to collect soil samples from just below each nesting hole. We restricted soil sample collection only in the case of occupied burrows that were used for breeding during our study period. We were unable to collect soil samples from areas without Sand Martin colonies because no other suitable nesting embankments were found in the study area.

The soil samples were dried for 1 month using blotting papers, and then 100 g of air-dried soil from each sample was sieved into 5 different sizes (1 mm, 0.5 mm, 0.425 mm, 0.106 mm, and 0.02 mm) using a Retsch AS 200 sieve shaker. In addition, pH and EC values were measured using a pH/Cond 340i and METTLER TOLEDO EasyFive Plus, respectively. A 1/2.5 ratio of soil/distilled water was used to measure these values. These variables could contribute to differences in the physical characteristics of nest holes.

### 2.4. Statistical Analyses

Because our data were hierarchically structured, i.e., the soil structure of individual nests in the same colony might be more similar than in individual nests located in different colonies, we used Linear Mixed Model Analyses to construct a two-level model (individual nest and colonial level) and null model (intercept-only model). The null model is normally used in hierarchical models to detect if the variable has a level (in our case, colonies) that significantly affects the intercept of the dependent variable at the lower level (in our case, nest hole). By using a Mixed Modelling approach, the variation in the dependent variable caused both the individual level (here, 80 individual holes) and group level (here, four colonies, Appendix A) was investigated. Through these analyses, we portioned variance and covariance into within-group and between-group components. By using the produced estimates of covariance parameters, residuals, and intercept (subject = colony), we calculated the intercorrelation by using the equation: group level/(individual level + group level). Secondly, we ran the same analyses by using a lakeshore variable as a group-level variable (two groups, colonies located in lakeshore areas and non-shore colonies; Appendix A). Thirdly, we used the Wald test to evaluate the significance of both the individual hole and colony-level variables (Appendix A). Finally, linear regression analyses were used to conduct a more detailed analysis of the effects of soil particle sizes, pH, EC, and lake site (used as a dummy variable; lakeshore colonies were coded as 1 and non-lakeshore colonies coded as 0; both groups had two colonies) on nest hole morphology (nest depth, distance between holes, width of the nest opening, and height of the nest opening).

A variance inflation factor (VIF) was used to measure the amount of multicollinearity in regression analysis. VIF values < 4.00 were considered representative of no multicollinearity. To determine that the residuals followed the normal distribution, we visually checked corresponding histograms and QQ plots. Normal P-P Plots of Regression Standardized Residuals were used to ensure that the relationships between dependent and explanatory variables were linear. The Durbin–Watson statistic was used to evaluate the possible autocorrelations in the residuals in the regression analyses; a value of 2 indicated no autocorrelation in the sample, values less than 2 indicated positive autocorrelations, and values > 2–4 indicated a negative autocorrelation. The Kruskal–Wallis H test was used to compare the four different colonies, and the Dunn post hoc test was used for pairwise multiple comparisons. We also evaluated relationships between individual variables using rho from Spearman’s correlation test. All tests were conducted by using the IBM SPSS statistical program Version 28.0.0.0.

## 3. Results

### 3.1. General Characteristics of Nesting Colonies

The four colonies had a total of 2510 burrows (mean = 627.0 ± 309. SD; range 278–851), of which 2305 (91.83%) were used for breeding during the study year. The use level of burrows in individual colonies was 91.92% (*n* = 458) in non-lakeshore site CV1, 90.64% (*n* = 278) in non-lakeshore site CV2, 90.98% in lakeshore site DMS, and 93.09% (*n* = 851) in lakeshore site DMC colony (Table 1). The height of the embankment varied between 2.4 and 3.2 m, the length of the embankments varied between 12 and 15 m, distance from the lakeshore varied between 0.015 km and 0.94 km, distance from the road ranged between 4.7 km and 5.83 km, and distance from the settlements varied between 0.22 km and 1.63 km (Table 1).

The mean (±SD) particle size was 0.123 mm ± 0.098 mm. The most common particle size class was 0.106 mm (Figure 3). The average pH of the samples was 8.81 (±0.30 SD), and the mean EC value was 171.00 (±67.27 SD).

### 3.2. Factors Influencing on Nest Characteristics

According to the results of the Linear Mixed Modelling, when comparing the impacts of both the colony (*n* = 4) and individual nests (*n* = 80) on the dependent variables (tunnel depth, distance between holes, and width and height of the entrance opening), a smaller proportion (2–33%) of the variance was explained by the grouping level (colony) factor than the individual level (nest) factor (67–98%; Appendix A). The results were basically the same when we conducted the same analyses using the location of nests (two groups, lakeshore vs. non-lakeshore colonies; *n* = 2 colonies in both groups; Appendix A). The proportion of variance of the dependent variables was mainly explained by the individual nests (from 67 to 98%; Appendix A). Moreover, based on the Wald tests, the higher-level intercepts of both the colony (Appendix A) and lakeshore (Appendix A) were non-significant. Therefore, further multilevel modeling was not necessary, and we used linear regressions in the subsequent analyses.

According to the linear regression analysis, tunnel depth was longer in nests with greater pH (Figure 4; Table 2) and EC (Table 2), and nests located in the lakeshore areas (mean = 55.90; SD = 18.65, *n* = 40) than in the non-lakeshore areas (mean = 51.33, SD = 11.80, *n* = 40; Table 2). Distance between nest holes increased with the EC (Table 2) and was shorter in nests located in the lakeshore areas (mean = 4.50, SD = 1.73, *n* = 40) than in the non-lakeshore areas (mean = 6.84, SD = 2.42, *n* = 40) (Table 2). The width of the entrance opening increased with soil particle size (Table 2) and was wider in nests located in the lakeshore areas (mean = 6.35, SD = 0.87, *n* = 40) than in non-lakeshore areas (mean = 5.65, SD = 0.79, *n* = 40; Table 2). No significant predictors were found for the height of the nest opening.

Tunnel depth, distance between tunnel holes, and width of the entrance opening differed between colonies (Table 3). Tunnel depth was greater in the holes in the DMS colony than in other colonies, and the holes were deeper in CV2 than in the DMC colony. Distance between nesting holes was greater in the CV1 colony than in the DMS or DMC colonies. The width of the entrance opening was larger in the DMS and DMC colonies than in the CV2 colony.

Tunnel depth correlated positively with pH (Spearman’s rho = 0.484, *p* < 0.001, *n* = 80) and EC (Spearman’s rho = 0.321, *p* = 0.004, *n* = 80; Appendix A). Distance between holes correlated negatively with EC (Spearman’s rho = −0.374, *p* < 0.001, *n* = 80, Appendix A). The width of the entrance hole correlated positively with EC (Spearman’s rho = 0.353, *p* = 0.001, *n* = 80, Appendix A). No significant correlations were detected between the nest characteristics (the tunnel depth, distance between the nest holes, width of the entrance opening, and height of the entrance opening; Spearman’s rho; *p* > 0.05, *n* = 80, Appendix A). There was a positive correlation between the pH and EC (Spearman’s rho = 0.458, *p* < 0.001, *n* = 80, Appendix A).

## 4. Discussion

The observed colony sizes varied between 278 and 851 holes, and the occupation level of the holes was as high as 92%. The mean soil particle size was 0.123 mm, and most particles belonged to the 0.106 mm size class. We found significant differences in the depth of tunnels, the distance between nest holes, and the width of entrance holes in Sand Martin nesting colonies. In general, tunnel depth was greater, and holes were located closer to one another in colonies located near the lakeshore than in colonies distant from the lake. Tunnel depth increased with the pH and EC. There was some indication that the distance of the nest holes increased with EC, and the width of the entrance opening increased with increasing particle size and was wider in nests in the lakeshore colonies than in non-shore colonies.

The observed mean colony size corresponded well with the colony sizes reported in Hungary (110–1200 [58]) or was greater than reported, as in Britain (e.g., mean 38 in Britain [21]), California, USA (mean = 269 [59]), or the Czech Republic (mean = 327 [24]). In our case, most of the holes in colonies were used for breeding during our study period, whereas other studies have reported breeding occupancy levels as low as 56% in Pennsylvania and Vermont, USA [60], 56% in California, USA [60], 63.82% in Hungary [59], and 58% in Finland [48]. However, it is difficult to compare the degree of occupation in such geographically distant regions without analyzing environmental factors. Differences in the occupancy rates of the Sand Martin nest tunnels in different regions may be influenced by a combination of environmental factors [61,62,63], anthropogenic effects [52,64], and population density [63,65]. Environmental conditions, such as climate [66], food availability [67], and habitat differences [66,68] can also influence nesting behavior [69]. In addition, human activities, such as urbanization [70], habitat destruction [71], and agriculture [72], can affect nesting success [69]. Higher population densities can lead to more competition for nesting sites [42]. Also, survey methods and techniques may vary, affecting the reported occupancy rates [69] referenced in our study. Understanding these factors is crucial to conservation and habitat management efforts for this species.

The tunnel depths in our current study (mean 53 cm; min: 25 cm and max: 103 cm) vary within similar ranges as they did in the data reported by Svensson [73] (mean 58 cm, min: 18 cm, and max: 90 cm, Ammarnäs-Sweden), Hopkins and Officer [74] (min: 35 cm and max: 103 cm), and Mondain-Monval and Sharp [75] (min: 30 cm and max: 100 cm, Whittington, Lancashire, United Kingdom). We assumed that differences in colony locations (lakeshore vs. river banks vs. artificial nest locations) could contribute to these differences.

Our results support the earlier finding that Sand Martins prefer soils in which particle size is less than 0.9 mm [28,29] However, earlier studies have not been highly detailed regarding smaller fractions than 0.9 mm particle sizes. Their influence needs to be studied in more detail. Our study showed that the mean particle size and most common particle size class was approximately 0.1 mm, suggesting that nests were primarily constructed in fine sand (silt). Spencer [60] found that the soil of 25 Sand Martin colonies comprised 50% or more of sand or fine gravel. Heneberg [29] indicated that a high penetrability resistance is an important factor for nest site selection of the Sand Martin, i.e., the Sand Martins will avoid sites with too compact or loose substrate for nesting to avoid nest collapses [25]. Yuan et al. [76] showed that Blue-tailed Bee-eaters prefer to nest in sandy loam and sandy clay loam soils rather than clay loam.

In general, soil pH determines the acidity and alkalinity level of the soil. Low pH levels (acidic soils) can promote tight binding of soil particles by increasing the soil’s hardness. In this case, soil penetration can be more difficult [77]. However, high pH levels (alkaline soils) can loosen soil particles and facilitate soil penetration. EC reflects the salt content in the soil [77]. High levels of EC can affect soil structure by keeping soil particles together, thereby increasing soil hardness [78]. This can make soil penetration more challenging. Conversely, low levels of EC allow soil particles to be looser with easier penetration [78]. The observation that tunnel depth increased with pH and EC may be related to the alkaline nature of the Lake Van shore.

The difference in nest tunnel depth between colonies may also be caused by the differences in the soil EC. It is possible that soil EC could affect the depth of nest holes in burrowing birds, although this would likely depend on the specific characteristics of the soil and the preferences of the bird species [35]. Soil EC is related to its water content, mineral composition, and organic matter content [36,37]. Soils with a high EC may be more prone to flooding or erosion [38], which could make them less stable for burrowing birds. In such cases, burrowing birds may choose to build their nest holes at a shallower depth to avoid these potential risks [39]. Similar to the findings of Evans [38], soils with a low EC may be more stable and less prone to flooding or erosion [40], which could allow burrowing birds to dig deeper nesting holes because of the greater stability of the soil [24].

It was observed that the nesting holes with small soil particle sizes (0.106 mm) were deeper than the other nesting holes (0.425 mm). Notably, it was observed that nesting holes with a smaller soil particle size (0.106 mm) exhibited greater depth compared to the other nesting holes (0.425 mm). Consistent with the values of our findings, Heneberg [79] also reported that smaller soil particles may be more prone to compaction and may be more difficult to excavate, which could result in deeper nest holes. However, larger soil particles may be easier to excavate but may be less stable and more prone to collapse, which could potentially impact the structural integrity and thermal properties of the nest [30].

The depth of a Sand Martin nesting tunnel is generally related to the height of the embankment or sand bank where the tunnel is located. Sand Martins typically dig their burrows into steep sand banks or cliffs, and the depth of the burrow is determined by the height of the embankment. In general, Sand Martins dig their burrows to a depth of about 60–90 cm below the surface of the sand bank [3]. However, the depth of the burrow can vary depending on the height of the embankment and the soil composition. In some cases, Sand Martins may dig their burrows to a deeper depth to reach more stable soil or provide additional protection from predators [80,81].

The size of the entrance opening can also be influenced by the need to control humidity and temperature within the nest [82]. A smaller entrance opening may be beneficial for regulating the internal environment of the nesting hole. In comparison, a larger entrance opening may be necessary to allow for the exchange of air and moisture with the surrounding environment. Sand Martins may construct wider nesting tunnel entrances in warmer, more humid environments to increase ventilation and reduce the risk of overheating.

Heneberg [79] found a statistically significant relationship between the width of the entrance opening and the distance to the bottom of the bank in Sand Martin nests. In contrast, Sieber [31] found strong correlations between the clay fraction of the soil (<0.02 mm in size) and the tunnel depth and entrance opening width in Sand Martin nests. However, we did not find statistically significant correlations between colonies in terms of tunnel depth, nesting entrance opening width, and tunnel height. In contrast, a moderate negative correlation was found between the CV-1 and CV-2 colonies in terms of distance between nests. However, multiple reasons might be predictive regarding the correlation coefficients. The correlation coefficients might be related to wide variation in nesting habits across bird species [83]; for this reason, it might not be possible to generalize common trends or relationships. Additionally, even if the same species were monitored, their relevant nesting habits might be influenced by various factors, such as environmental conditions, availability of nesting sites, and predator pressures, which may vary widely across different habitats and regions [7,83,84,85]. Such variations, in turn, might lead to different nesting strategies evolving in different bird populations, further complicating attempts to find correlations between nesting habits [7,10]. Of the critical factors considered for such analyses, the physiology, behavior, and genetics of the birds might also be considered to understand their nesting habits [7]. Therefore, the present findings cannot be dedicated to the differing distances of all bird species and habitats from the lakeshore. In this context, each burrowing bird species should be evaluated in its distribution areas.

Several factors can influence the structure and dimensions of nests [8,83]. Research has shown that the depth and spacing of nest holes and the dimensions of the entrance can have a significant effect on passerine bird species nesting behavior [4,85], as observed in our study of Sand Martins as well as in other studies [3,19,24] and other burrowing species [6,7]. In this regard, numerous reports have revealed that the factors above significantly affect the location and spacing of nesting holes, the depth of the nests, and the dimensions of the nest entrance in general [8,10,11] and in Sand Martin nesting colonies in particular [3,24,28,42,86]. These factors include the availability of suitable nesting sites [87,88], the size of the nesting colony [89], and the availability of resources such as food and water [90]. Sand Martins are known to be territorial and may defend their territory, including their nesting site, from other Sand Martins or burrowing bird species [7].

Significant differences in nest hole depth were found between the CV-1 and DMS colonies but not between the others. Sand Martins face several potential nest predators, including mammals such as foxes, minks, and weasels (*Mustela* sp.) and birds such as crows (*Corvus* sp.) and other corvids [43,85,91,92,93,94,95,96]. The DMS and DMC colonies have burrows dug by red foxes at the bottom of the embankments, whereas no such predation risk was detected in the CV-1 and CV-2 colonies. Predator pressure may influence the depth of nest tunnels, protecting them from predation and potential nest damage [28]. In the case of burrowing birds, the depth of their nest tunnels can significantly affect the amount of natural light that enters the nest chamber [97]. Deeper tunnels receive less natural light, which can have several consequences. First, it reduces the visibility of the nest chamber, making it more difficult for visual predators and parasites to locate and access nests. Second, reduced light levels may discourage certain parasites or insects that require light for activity or navigation [27,97,98]. Therefore, understanding the relationship between tunnel depth and the amount of light in the nest chamber may provide insights into the factors that influence parasite detection and predation risk for these burrowing bird species [99]. Other factors influencing burrow depth include the availability of nesting material, competition for nest sites, and substrate characteristics [3,24,28,29,100,101].

Deeper burrows have been observed in Sand Martin colonies as a defense strategy against red fox predation, particularly in the DMS and DMC colonies [92]. This was not observed in the CV-1 and CV-2 colonies. The presence of predatory species and wider, well-excavated tunnels in the DMS and DMC colonies may be related to differences in the width of nest tunnel entrances.

Sand Martins may be more likely to nest closer together if suitable nest sites are limited in number or if they are in high demand. Concerning distances between nesting holes, the nesting holes at CV-1 and CV-2 were farther apart than those of DMS and DMC. The DMS and DMC colonies were located close to the lakeshore, with a shorter length of embankments and a higher number of nests in relation to CV-1 and CV-2. Thus, our results support the view that Sand Martins favor nest sites near water bodies [1,46,53]. The limited embankment surface area and access to food resources partly explain the shorter distance between the nesting holes of DMS and DMC. Understandably, the closer proximity to the lakeshore of the DMS and DMC colonies may provide easier access to aquatic resources such as invertebrates, which could affect the behavior and nesting hole location of the individuals in these colonies.

The location of the colony and the presence of anthropogenic pressure in the CV-1 and CV-2 colonies may also have impacts on the habitat quality and resource availability in these colonies (see also Richner and Heeb [102], which in turn manifested in the low number of nests and populations in the study region. When sufficient areas for nesting and resources are abundant, the nests may be spaced farther apart as the birds have more options for foraging and do not need to be as close to each other [20]. On the other hand, if resources are scarce, the nesting holes may be built closer together to pool the birds’ resources and increase the chances of survival for the entire colony [28]. Numerous reports have highlighted the influence of nesting behavior on the spacing of Sand Martin nest holes [27,41,103]. However, we observed closely spaced nest holes in the region (DMS and DMC colonies). We attribute the high number of nest holes in Sand Martin colonies (DMS and DMC) to their proximity to water sources and abundant food, mainly insects. In areas where the needs of Sand Martins are well met, with ample food resources and proximity to water sources, they show higher nesting activity. Corresponding to the abundance of resources, an increased number of nesting Sand Martins was observed in the region, but the embankment lengths of these nesting hole locations were short. This might force birds to nest more closely to each other to accommodate all the birds. However, nesting behavior in the context of high resource abundance but limited nesting locations deserves more in-depth study. Further research into the breeding success of Sand Martins nesting on different types of banks and sand is also needed.

## 5. Conclusions

Our results indicated that the location of the colony was a critical predictor of the physical properties of the nesting holes, including tunnel depth and space between tunnel holes. Sand Martins build deeper nest tunnels near the water sources and soils with greater pH and electronic conductivity values. Nest holes were located nearer to each other in colonies located at the lakeshore than further away, and between-hole distances increased with the EC. Our results indicated that Sand Martins would avoid sites with too compact or loose soils for their nesting, probably to avoid nest collapses. Vertical lakeshore embankments offer good nesting sites for the Sand Martins, which should, therefore, be protected. Future research could expand on these findings by investigating the impact of other environmental factors on the nesting habits of Sand Martins, such as temperature, vegetation cover, and prey availability. Also, we should have collected soil samples from unoccupied, unused nesting burrows. Further investigations are required to determine possible differences between occupied and unoccupied holes. Additionally, more studies are needed to understand the long-term population trends and distribution patterns of Sand Martins across different regions. Masoero et al. [104] have suggested that appropriate gravel management can help mitigate the decline of the Sand Martin populations. Our results will shed light on the Sand Martins’ adaptability and the potential impact of human activities on their nest site selection. Our findings can be used in conservation strategies, such as habitat management and the creation of artificial nesting sites or even nests, to support the occurrence and success of the Sand Martin populations in diverse areas (see also [45]).

## Figures and Tables

**Figure 1 animals-13-03463-f001:**
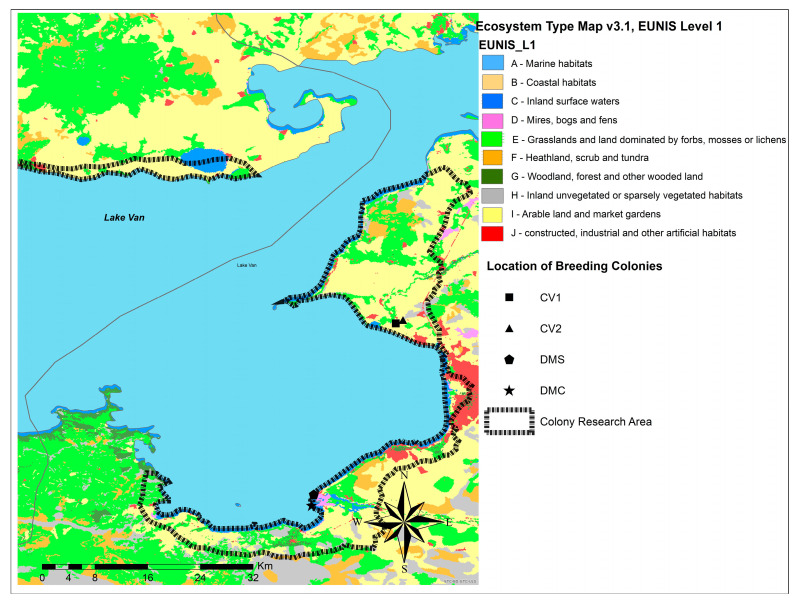
Habitat map of Sand Martin nesting colonies in the Lake Van Basin (this map was generated using ArcMap 10.2 and obtained from the ArcGIS online database).

**Figure 2 animals-13-03463-f002:**
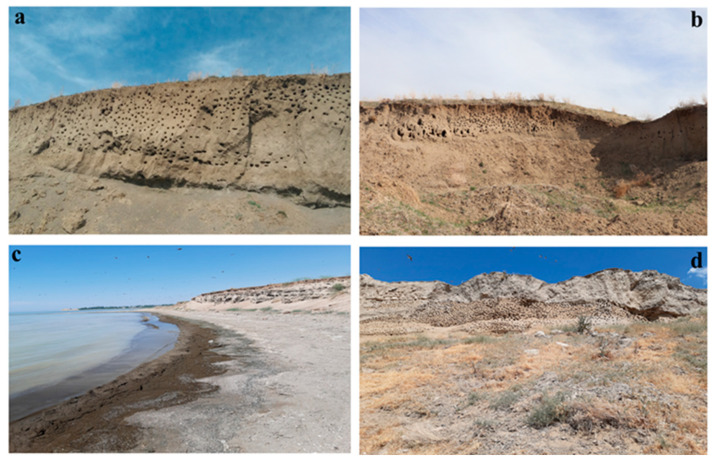
Breeding sites of Sand Martins in the study area. (**a**) = Çitoren Village (CV1), (**b**) = Çitoren Village (CV2), (**c**) = Dilkaya Mound Coast (DMC), and (**d**) = Dilkaya Mound School (DMS). (Photos by EC). Colonies at the Çitoren Village (**a**,**b**) are located in non-shore areas, whereas colonies at the Dilkaya Mound are located along lakeshore areas.

**Figure 3 animals-13-03463-f003:**
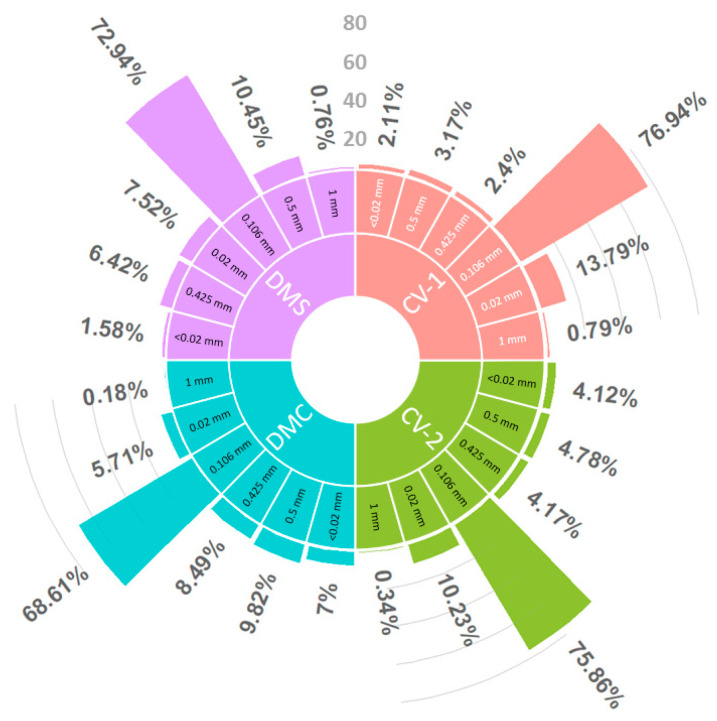
Particle size (mm) distribution (%) in different Sand Martin colonies (non-lakeshore CV-1 = Çitoren Village (CV1), non-lakeshore CV-2 = Çitoren Village (CV2), lakeshore site DMC = Dilkaya Mound Coast, and lakeshore site DMS = Dilkaya Mound School). Particles were separated using a Retsch AS 200 sieve shaker.

**Figure 4 animals-13-03463-f004:**
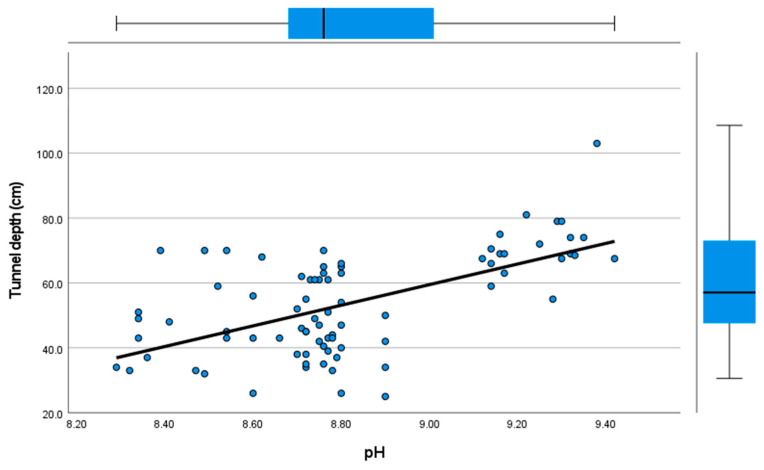
Relationship between pH and tunnel depth of Sand Martin nests in Turkey (*n* = 80).

**Table 1 animals-13-03463-t001:** Basic characteristics of the different Sand Martin nest colonies (non-lakeshore site CV-1 = Çitoren Village (CV1), non-lakeshore site CV-2 = Çitoren Village (CV2), lakeshore site DMC = Dilkaya Mound Coast, and lakeshore site DMS = Dilkaya Mound School).

Region	HE (cm)	LE (m)	NN (n)	DW (km)	DH (km)	DS (km)
CV-1	320	22	458	0.94	5.8	1.6
CV-2	275	12	278	0.9	5.83	1.63
DMS	240	8	920	0.03	4.16	0.22
DMC	260	15	854	0.015	4.7	1.65

HE: height of embankment; LE: length of embankment; NN: number of nests; DW: distance to lakeshore; DH: distance to highway; and DS: distance to settlements.

**Table 2 animals-13-03463-t002:** Results of the linear regression analyses related to the relationships between nest characteristics (tunnel depth, distance between the nest holes, and width of the entrance opening) and pH, EC (electrical conductivity), particle size, and location of the nest (lakeshore colony; non-lakeshore colony). Only significant results and variables are shown. SE = standard error. VIF = variance inflation factor (VIF) is a measure of the amount of multicollinearity in regression analysis.

	β	SE	t	P	VIF
Tunnel depth (R^2^ = 0.54; Adj. R^2^ = 0.51; SE = 0.51; F = 21.28, df = 4, 75; *p* < 0.001)					
Constant	−202.86	69.32	−2.93	0.005	
pH	27.72	8.38	3.30	0.001	3.92
EC	0.116	0.03	3.42	0.001	3.43
Location	−15.79	3.35	4.71	<0.001	1.87
Particle size	0.001	0.01	0.05	0.962	1.07
Distance between tunnel holes (R^2^ = 0.28; Adj. R^2^ = 0.43; SE = 2.09; F = 7.37, df = 4, 75; *p* < 0.001)					
Constant	25.00	13.20	1.90	0.062	
EC	0.01	0.01	2.04	0.045	3.43
Location	−2.53	0.64	3.96	<0.001	1.87
pH	−2.30	1.60	−1.44	0.155	3.92
Particle size	−0.001	0.002	−0.33	0.742	1.02
Width of the entrance opening (R^2^ = 0.26; Adj. R^2^ = 0.22; SE = 0.80; F = 6.49, df = 4;75; *p* < 0.001)					
Constant	15.35	5.04	3.05	0.003	
Particle size	0.002	0.001	2.29	0.025	1.11
Location	0.09	0.24	3.52	<0.001	1.87
pH	−1.21	0.61	−1.99	0.051	3.92
EC	0.004	0.002	1.49	0.139	3.43

**Table 3 animals-13-03463-t003:** Differences in the tunnel depth (cm), distance between tunnel holes (cm), width of the entrance opening (cm), and height of the entrance openings (cm) between the four Sand Martin breeding colonies (CV-1 = Çitoren Village (CV1), CV-2 = Çitoren Village (CV2), DMC = Dilkaya Mound Coast, and DMS = Dilkaya Mound School) in Turkey. Mean and SD values are given. Kruskal–Wallis H test and pairwise post hoc Dunn tests were used in statistical analyses. * Tunnel depth of individual burrows varied between 25 and 103 cm.

	Tunnel Depth *	Distance Between Tunnel Holes	Width of the Entrance Opening	Height of the Entrance Opening
Colony				
CV-1 (non-shore)	46.90 ± 11.91	7.60 ± 2.10	5.90 ± 0.88	3.40 ± 0.50
CV-2 (non-shore)	55.70 ± 9.70	6.07 ± 2.51	5.40 ± 0.56	3.60 ± 0.73
DMS (shore-site)	71.42 ± 9.80	4.87 ± 1.53	6.32 ± 0.76	3.75 ± 0.57
DMC (shore-site)	40.37 ± 10.51	4.12 ± 1.86	6.37 ± 0.98	3.82 ± 0.54
Kruskal–Wallis H	44.64	23.26	16.28	3.47
Df	3	3	3	3
Asymp. Sig.	0.001	0.001	0.001	0.325
Pairwise adjusted significant differences, Dunn test	CV2 > DMC	CV1 > DMC	DMS > CV2	not tested
	DMS > DMC	CV1 > DMS	DMC > CV2	
	DMS > CV1			
	DMS > CV2			

## Data Availability

The data presented in this study are available in Appendix A.

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
