# Peer review of "Sand-Related Factors Influencing Nest Burrowing Potential of the Sand Martins"

_animals, 2023, doi:10.3390/ani13223463_

Round 1
Reviewer 1 Report (New Reviewer)
Comments and Suggestions for Authors
12: could simply be “influencing the nesting locations of…”
13: “of four Sand Martin colonies,…”
27: Either “for the Sand Martin” or “for Sand Martins”.
65: in large colonies instead of groups
75-76: missing period at the end of the sentence. Also suggest giving a global context to “widely”, in the country? Hemisphere? Worldwide?
86. Missing period
102: Is it a consequence or a benefit for sand martins?
192: suggest changing to “all continents, except Antarctica”.
192: suggest changing to: “They are commonly found near bodies of water such as rivers, streams, and lakes in their distributed continents, however, they do not breed in Africa and South America.
210: Suggest a time frame, from 2016 -???
217: Were field surveys done from 2016 – 2021? If these are different survey types it should be clearer.
247: Could these distances be better visualized and compared in a small table?
265: Depended should be dependent
312-313. period missing in previous sentence. Final sentence doesn’t seem to belong?
314: There are two headings that are 2.5 Statistical Analysis – should these be different?
340: Don’t think it’s necessary to include sentence “Basic characteristics of individual colonies are shown in Table 1.”, that’s what the reference is for in the sentence after.
357: Axis on graph should be labelled.
367 & 368: 2% to 33%, is this a range or the variance went from 2% to 33%, I would change the ‘to’ to a dash if it is a range. Same goes for the individual level nest.
394 & 398: Make sure axis titles are all standardized (i.e. caps on each letter).
449: "nearer one another" sounds a bit strange grammatically, would suggest "closer to one another".
458: change to "during our study period"
464-468: Are there any references that could be added for these statements?
471: Suggest you change range to include using terms min and max, similar to the examples later on in the sentence.
Very interesting study, I really enjoyed reading this paper and learning more about Riparia Riparia!
Comments on the Quality of English LanguageGood! Few comments made above to improve clarity in some locations
Author Response
Reviewer 1
We thank the reviewer for the valuable comments for our manuscript. We have practically taken care all of them, but we wanted to notice than we have received comments from exceptionally many (five!) reviewers for our ms. Therefore, we have been forced to consider, at some level, the remarks of the all five reviewers at the same time. Our responses for your remarks are given in bold/italics after your comments.
12: could simply be “influencing the nesting locations of…”
We would keep this sentence as it now because Wiley English language consultant suggested to write it so. It is good to avoid to have two “-ing” (influencing and nesting) words so close each other’s.
13: “of four Sand Martin colonies,…”
Edited as suggested.
27: Either “for the Sand Martin” or “for Sand Martins”.
Edited as “for the Sand Martin”
65: in large colonies instead of groups
This sentence is also checked by the Wiley language consultant and therefore we will keep it as it is now. Note, that the word “colonial” was already mentioned at beginning part of the sentence: “Sand Martins are colonial burrowing nesters that build their nests in large groups”; i.e. it good to avoid use “colonial” twice in the same sentence.
75-76: missing period at the end of the sentence. Also suggest giving a global context to “widely”, in the country? Hemisphere? Worldwide?
Edited as: “Current studies indicate that Sand Martin populations have declined over the last decades e.g. in Finland [49] and North America [50,51].”
- Missing period
The study was conducted on 1997-1998, but we think that this information is reductant in the sentence: “Furthermore, the prevalence of the blood parasite, Trypanosoma avium in Bank Swal-lows has been as high as 50 % in Alaska [39].” Note, that this part of text is now moved from the Introduction to the Discussion section.
102: Is it a consequence or a benefit for sand martins?
The consequences are described in the following sentences: “Deeper tunnels receive less natural light, which can have several consequences. First, it reduces the visibility of the nest chamber, making it more difficult for visual predators and parasites to locate and access nests. Second, reduced light levels may discourage certain parasites or insects that require light for activity or navigation [50].” Note, that this part of text is now moved from the Introduction to the Discussion section.
192: suggest changing to “all continents, except Antarctica”.
Edited as suggested.
192: suggest changing to: “They are commonly found near bodies of water such as rivers, streams, and lakes in their distributed continents, however, they do not breed in Africa and South America.
This sentence has been now removed due to the suggestion of another reviewer.
210: Suggest a time frame, from 2016 -???
Edited as: .. from 2016 to 2021.
217: Were field surveys done from 2016 – 2021? If these are different survey types it should be clearer.
We have now clarified this like: “We systematically searched Sand Martin nesting colonies in the Lake Van study area (5.55 km2; Figures 1, 2) from 2016 to 2021”; “More detailed studies than searching colonies were conducted during daytime and evening hours between April 20 and September 23, 2021”
247: Could these distances be better visualized and compared in a small table?
This part of the text is now deleted because this information is given in the Table 1.
265: Depended should be dependent
The language of this sentence was corrected by the Wiley language consultant, so we want to keep here “was depended”. Note that this sentence is now removed to the Result section.
312-313. period missing in previous sentence. Final sentence doesn’t seem to belong?
Sorry about this error which has made by the managers in the Editorial office of the Animals. We have now corrected this.
314: There are two headings that are 2.5 Statistical Analysis – should these be different?
Sorry about this error which has made by the managers in the Editorial office of the Animals. We have now corrected this.
340: Don’t think it’s necessary to include sentence “Basic characteristics of individual colonies are shown in Table 1.”, that’s what the reference is for in the sentence after.
Agree, this sentence is now deleted.
357: Axis on graph should be labelled.
Thank you for highlighting this, we have formulated a totally new figure 3 which is in our mind clearer than the earlier one.
367 & 368: 2% to 33%, is this a range or the variance went from 2% to 33%, I would change the ‘to’ to a dash if it is a range. Same goes for the individual level nest.
Edited as suggested.
394 & 398: Make sure axis titles are all standardized (i.e. caps on each letter).
We have now used a standard way to present axis titles. Note, that due to the request of the Academic Editor, we have deleted away all the earlier bar figures, and given the corresponding information (means, SDs and n in the main text).
449: "nearer one another" sounds a bit strange grammatically, would suggest "closer to one another".
Edited as suggested.
458: change to "during our study period"
Edited as suggested.
464-468: Are there any references that could be added for these statements?
We have added references to support these statements.
471: Suggest you change range to include using terms min and max, similar to the examples later on in the sentence.
Changed as suggested.
Very interesting study, I really enjoyed reading this paper and learning more about Riparia Riparia!
Thank you for your very positive views towards our ms.
We have used Wiley's professional English editing services to correct our English usage, and we have done the minor corrections suggested by the reviewers.

Reviewer 2 Report (Previous Reviewer 2)
Comments and Suggestions for Authors
The paper has been partially corrected, apparently, it can be seen in the statistical methods and results. Nevertheless, unnecessary topisc in introduction and methods sections, as well as in Disscission, should be removed. The number of literature items cited is far too long and should be shortened.

English has been greatly improved
Author Response
Reviewer 2
We thank the reviewer for the valuable comments for our manuscript. We have practically taken care all of them, but we wanted to notice than we have received comments from exceptionally many (five!) reviewers for our ms. Therefore, we have been forced to consider, at some level, the remarks of the all five reviewers at the same time. Our responses for your remarks are given in bold/italics after your comments.
The paper has been partially corrected, apparently, it can be seen in the statistical methods and results. Nevertheless, unnecessary topics in introduction and methods sections, as well as in Discussion, should be removed. The number of literature items cited is far too long and should be shortened.
We thank the reviewer for her/his great input for our manuscript. We want to point out that we have already shortened the length of the ms (especially the Introduction and Discussion) considerable as compared for the original ms version. The method section is enlarged because the reviewer(s) requested some additional analyses, which we were forced to describe in our statistical analyses section. Also, as an e-Journal only, the Animals does neither has any length restrictions related the length of the ms nor the number of references (in fact, there shortened least 40 references in article type papers). However, we have now done our best to condense our writing style. For example, we have either removed, condensed or re-placed (from the introduction to the discussion) the following paragraphs/sentences that the reviewer suggested:
“Nesting strategies of birds are influenced by many environmental factors, both biotic and abiotic [1,2]”.
àdeleted
“Nest holes located higher on the bank are less vulnerable to ground predators, such as weasels (Mustela nivalis) or small mammalian predators (such as the woolly mouse opossum Micoureus cinereus, brown rat Rattus norvegicus, red-tail squirrel Sciurus granatensis, and spiny pocket mice Heteromys spp.) that can easily access nests located at lower heights [29-31]. Additionally, nesting in holes or burrows can provide greater protection against brood parasitism by the Cuckoo (Cuculus canorus) [32-37]. Ectoparasitism may be costly for the Sand Martin [38]. Furthermore, the prevalence of the blood parasite, Trypanosoma avium in Bank Swallows has been as high as 50 % in Alaska [39].”
àdeleted
“Parasites, including some ectoparasites and nest-dwelling insects, often rely on visual cues or phototaxis (movement towards or away from light) to locate their hosts or breeding sites [48,49]. In the case of burrowing birds, the depth of their nest tunnels can significantly affect the amount of natural light that enters the nest chamber [50]. Deeper tunnels receive less natural light, which can have several consequences. First, it reduces the visibility of the nest chamber, making it more difficult for visual predators and parasites to locate and access nests. Second, reduced light levels may discourage certain parasites or insects that require light for activity or navigation [50]. Therefore, understanding the relationship between tunnel depth and the amount of light in the nest chamber may provide insights into the factors that influence parasite detection and predation risk for these burrowing bird species [51].”
àdeleted
“In general, soil pH determines the acidity and alkalinity level of the soil. Low pH levels (acidic soils) can promote tight binding of soil particles, by increasing the soil hardness. In this case, soil penetration can be more difficult [64]. However, high pH levels (alkaline soils) can loosen soil particles and facilitate soil penetration. EC reflects the salt content in the soil [64]. High levels of EC can affect soil structure by keeping soil particles together, thereby increasing soil hardness [65]. This can make soil penetration more challenging. Conversely, low levels of EC allow soil particles to be looser with easier penetration [65].”
à replaced in the Discussion
“We hypothesized that soil structure and proximity to water will influence the mor-phometric characteristics of nesting burrows. We predicted that nests closer to water sources would have deeper nesting holes, wider and higher nest entrances, and shorter distances between nesting holes. Additionally, we hypothesized that soil pH, EC, and soil particle size, can influence the morphometric characteristics of Sand Martin nests.”
à replaced at the end of the Introduction.
“They are commonly found near bodies of water such as rivers, streams, and lakes in North America, Europe, Asia, Africa, and South America. However, they do not breed in Africa and South America.”
àdeleted.
“The non-shore CV1 colony was 0.94 km from the nearest waterbody, 1.6 km from the nearest settlement, and 5.8 km from the nearest highway. The non-shore CV2 colony was 0.90 km from the closest waterbody, 1.64 km from the nearest settlement, and 5.84 km from the nearest highway. Shore-site DMS colony was 0.03 km from the nearest waterbody, 0.22 km from the nearest settlement, and 4.16 km from the nearest high-way. The shore-site DMC colony was 0.015 km from the closest waterbody, 1.7 km from the nearest settlement, and 4.7 km from the nearest highway.”
àdeleted because the same information is given in the Table 1.
“The spacing between sampled holes was depended by the length of each colony's embankment: approximately 1 m for CV-1, 0.60 m for CV-2, 0.40 m for DMS, and 0.75 m for DMC.”
àwe want to keep this in the chapter 2.4. Soil Sampling chapter and not move it for the Results section, because this information describes here how the soil sampling was conducted. I.e. what was the distance between tunnels from which soil samples were taken in the individual colonies, and why the distance varied a little bit between colonies.
Comments on the Quality of English Language
English has been greatly improved.
We have used Wiley's professional English editing services to correct our English usage, and we have done the minor corrections suggested by the reviewers.

Reviewer 3 Report (New Reviewer)
Comments and Suggestions for Authors
This manuscript went already under extensive revisions. It is overall good but there are some major parts that require some clarification before this can be accepted:
- there is mismatch between dates. When are the surveys done? Why do you state that you did the surveys since 2016 and then you state that surveys were only done 20-23 Sept 2021? This is a very limited sampling period.
- Something wrong with lines 312-313
- why VIF<6? There is no reference. Usually it is <4
- I understand you used QQ plots but there seems to be something weird with model outputs. Fig 7 does not really indicate significant differences (means and 95%CI of the two groups are overlapping), so probably your dependent variable does not have the best fit family and you should use generalised linear mixed models (ideally using R)
- Numbers in fig 5 should have . instead of . for decimals. also missing y axis label
- Table 2. You should report all the results as it is important to show you did not include multicollinear factors. Also it is the norm to add all parameters, and you just indicate significant results
Comments on the Quality of English LanguageNo major issues, but manuscript is a bit long and the message is diluted, difficult to find the main information and findings and why they are relevant.
Author Response
We thank the reviewer for the valuable comments for our manuscript. We have practically taken care all of them, but we wanted to notice than we have received comments from exceptionally many (five!) reviewers for our ms. Therefore, we have been forced to consider, at some level, the remarks of the all five reviewers at the same time. Our responses for your remarks are given in bold/italics after your comments.
Reviewer 3
This manuscript went already under extensive revisions. It is overall good but there are some major parts that require some clarification before this can be accepted:
- there is mismatch between dates. When are the surveys done? Why do you state that you did the surveys since 2016 and then you state that surveys were only done 20-23 Sept 2021? This is a very limited sampling period.
As stated in the manuscript, we have studied the Sand Martins in the region since 2016 (e.g. searching the colonies), but the main field studies, like the soil sampling, were conducted on 20-23 Sept 2021. We have clarified now this like: “We systematically searched Sand Martin nesting colonies in the Lake Van study area (5.55 km2; Figures 1, 2) from 2016 to 2021”; “More detailed studies than searching colonies were conducted during daytime and evening hours between April 20 and September 23, 2021”
- Something wrong with lines 312-313
Sorry about these errors (e.g. duplicate subtitles and one strange sentences) which has made by the managers in the Editorial office of the Animals. We have now corrected these errors; i.e. there is only one 2.5. Statistical Analyses chapter.
- why VIF<6? There is no reference. Usually it is <4
Agreed. This was a misprint, the cut value used was 4 (see Table 4). Thank you for highlighting this.
- I understand you used QQ plots but there seems to be something weird with model outputs. Fig 7 does not really indicate significant differences (means and 95%CI of the two groups are overlapping), so probably your dependent variable does not have the best fit family and you should use generalized linear mixed models (ideally using R)
Thank you pointing out this topic. According to the Linear regression modelling analysis, location of the colony has an impact on the tunnel dept (Table 2), but as you rightly indicated , 95%CI values of the two groups (shore-sites and non-shore sites) overlapped. That was partly due the fact that one shore-site colony had the longest, and one them the shortest tunnel depth (see the Table 3). To avoid confusions, we have now deleted the Fig. 7, the readers can see the site species differences more clearly from the Table 3.
In fact, we have used firstly the Linear Mixed Modeling approach for our data (see lines 318-322). However, based on their results (Wald tests), the higher-level intercepts of both the colony (Table S1) and lakeshore (Table S2) were non-significant. Therefore, further multilevel modelling was not necessary, and thereby the use of linear regressions in the subsequent analyses were satisfied (see lines 324-329). By following this approach, also the interpretation of the results are much more simple.
We are not users of the R-program, but the fortunately the SPSS -program has all the tools that we needed.
- Numbers in fig 5 should have . instead of . for decimals. also missing y axis label
Thank you for highlighting the missing axis label names. Because of the request of the Academic Editor, we have deleted away all the original bar figures, and given necessary information in the main text.
- Table 2. You should report all the results as it is important to show you did not include multicollinear factors. Also it is the norm to add all parameters, and you just indicate significant results.
Done.
Comments on the Quality of English Language
No major issues, but manuscript is a bit long and the message is diluted, difficult to find the main information and findings and why they are relevant.
We have used Wiley's professional English editing services to correct our English usage, and we have done the minor corrections suggested by the reviewers.
We want to point out that we have already shortened the length of the ms (especially the Introduction and Discussion) considerable as compared for the original ms version. The method section is enlarged because the reviewer(s) requested some additional analyses, which we were forced to describe in our statistical analyses section. Also, as an only e-Journal, the Animals does neither has any length restrictions related the length of the ms nor the number of references (in fact, there shortened least 40 references in article type papers). However, we have now done our best to condense our writing style.

Reviewer 4 Report (New Reviewer)
Comments and Suggestions for Authors
When I first saw the manuscript, I was confused why it was presented with some comment in correction mode. Was it already reviewed for the first time?
I was also confused about reading pH and electrical conductivity in relation to birds. My main concern is, that the paper is based on 4 sites, and they also seem to be spatial related. Even when the sample size of the nesting tubes is 8 the sample size in general is reduced to 4, especially when the authors talk about the factor distance from water. With this design it is not clear – is the site or is it the distance to water – which effects the nesting tubes.
Introduction
Here the authors mix up birds using borrows and some digging tubes ore holes by themselves. Form me it is also critical to mix up European and American species, songbirds and non-songbirds. In this case the introduction should by straighter.
The sand martin as a species is described in the introduction and the methods. This is redundant.
The introduction should have a clear structure – consistency of the river bank, predation, brood parasitism, parasites. In the introduction the different aspects are described quite deeply and should shifted to the discussion.
Research questions should be presented at the end of the introduction all together and should be shortened.
Methodes
The two blue/grey maps in figure 1 make no sense. The same map presented in Fig 2 should also be available for the non-shore colonies. In the first part it is already described that the non-shore colonies underlay the same conditions. Is does not make sense to me to consider them as separated.
Line 246-259 Describe data in table 1, this is redundant.
The paragraph statistical analysis is double
I find it very confusing that the nest site characteristics are related to pH, particle size and electrical conductivity. The birds are not selecting this variables, so name them after what they stand for.
I think the consumption “distance from the lake was a critical predictor of the 620 physical properties of the nesting holes” is not allowed. The reason I explained before.
The manuscript is very long and explanations are quit long an complicated. The authors should focus on the main message of the paper. The very intensive statistical analysis it not necessary, sometimes misleading and confusing. If size of colony and occupancy rate will be presented in this paper, it should be considered in the title.
Author Response
We thank the reviewer for the valuable comments for our manuscript. We have practically taken care all of them, but we wanted to notice than we have received comments from exceptionally many (five!) reviewers for our ms. Therefore, we have been forced to consider, at some level, the remarks of the all five reviewers at the same time. Our responses for your remarks are given in bold/italics after your comments.
Reviewer 4
When I first saw the manuscript, I was confused why it was presented with some comment in correction mode. Was it already reviewed for the first time?
You are right, our manuscript has already gone thought one review round, and those track-and-change correction marks were unfortunately leftovers of this round due the changes made by the editorial office of the Animals for our initial submission. We are sorry about this.
I was also confused about reading pH and electrical conductivity in relation to birds.
Both the pH and electrical conductivity can influence on the soil structure, and thereby on burrowing possibilities of the Sand Martins. Also, for the researchers, conservationists and land use managers, it would be quite easy to measure the pH and electrical conductivity of the sand banks, and thereby evaluate the suitability of different Sand banks for the use (burrowing the tunnels) of the Sand Martins.
My main concern is, that the paper is based on 4 sites, and they also seem to be spatial related. Even when the sample size of the nesting tubes is 8 the sample size in general is reduced to 4, especially when the authors talk about the factor distance from water. With this design it is not clear – is the site or is it the distance to water – which effects the nesting tubes.
We agree that the total number of colonies is low. Unfortunately, even our intensive multi-year colony search work with the Sand Martin, there are no other colonies in our study region. We have sampled a total of 80 burrows (20 burrows in each four colonies), not 8. According to our Linear Mixed Modeling analyses, different burrows (n = 80) can be used as independent observations because the soil structure of individual nests in the same colony were NOT more similar than in individual nests located in different colonies.
We agree that it better to speak about the site(s) (colonies) instead directly speaking about the distance to the water. It is important if the colony is either located at the lake shore or not. We have edited the ms correspondingly when appropriate, e.g. we have edited the title of the ms.
Introduction
Here the authors mix up birds using borrows and some digging tubes ore holes by themselves. Form me it is also critical to mix up European and American species, songbirds and non-songbirds. In this case the introduction should by straighter.
Because there are only a few bird species that use burrows in lands for their nesting, in our mind, it would be good to mentioned these species shortly. And moreover, there is quite a few publications that have been conducted related to the burrowing bird species. We have already skipped burrowing maritime species away from our Introduction section. Therefore, we want to keep citations currently presented in our Introduction section. However, we have tried to condense the Introduction section a little bit.
The sand martin as a species is described in the introduction and the methods. This is redundant.
Agreed, we have deleted the separate chapter 2.2. Study Species, and moved some parts of it into the Introduction.
The introduction should have a clear structure – consistency of the river bank, predation, brood parasitism, parasites. In the introduction the different aspects are described quite deeply and should shifted to the discussion.
We have put more attention to the structure of the Introduciuton, and either deleted or moved some parts of it for the Discussion section as suggested by the other reviewers.
Research questions should be presented at the end of the introduction all together and should be shortened.
We have now presented our study aims, hypotheses and predictions at the end of the Introduction a more condense way.
Methodes
The two blue/grey maps in figure 1 make no sense. The same map presented in Fig 2 should also be available for the non-shore colonies. In the first part it is already described that the non-shore colonies underlay the same conditions. Is does not make sense to me to consider them as separated.
Agreed. We have now deleted the old Figure 1, enlarged the scale of the new Figure 1 and added the study area boarders for the Figure 1.
Line 246-259 Describe data in table 1, this is redundant.
Agreed. We have now deleted from the text the duplicated information given in the Table 1.
The paragraph statistical analysis is double
Sorry about these errors (e.g. duplicate subtitles and one strange sentences) which has made by the managers in the Editorial office of the Animals. We have now corrected these errors; i.e. there is now only one 2.4. Statistical Analyses chapter.
I find it very confusing that the nest site characteristics are related to pH, particle size and electrical conductivity. The birds are not selecting this variables, so name them after what they stand for.
We are studying factors influencing on BURROWING possibilities of the Sand Martin. And burrowing possibilities of the sand impacts on which kinds of burrows the birds are excavating. Measuring the particle size is commonly used in these kinds of studies [1,3,29-36], and also the pH and electrical conductivity [25,41]. All of these variables can influence on the soil structure, and thereby on burrowing possibilities of the Sand Martins. Also, for the researchers, conservationists and land use managers, it would be quite easy to measure the pH and electrical conductivity of the sand banks, and thereby evaluate the suitability of different Sand banks for the use (burrowing the tunnels) of the Sand Martins. Because we have measured particle size, pH and electrical conductivity, we should the actual names of these variables.
I think the consumption “distance from the lake was a critical predictor of the 620 physical properties of the nesting holes” is not allowed. The reason I explained before.
We agree that it is better to speak about the site(s) (colonies) instead directly speaking about the distance to the water. It is important if the colony is either located at the lake shore or not. We have edited the ms correspondingly when appropriate.
The manuscript is very long and explanations are quit long an complicated. The authors should focus on the main message of the paper. The very intensive statistical analysis it not necessary, sometimes misleading and confusing. If size of colony and occupancy rate will be presented in this paper, it should be considered in the title
We thank the reviewer for her/his great input for our manuscript. We want to point out that we have already shortened the length of the ms (especially the Introduction and Discussion) considerable as compared for the original ms version.
The method section is enlarged because some reviewer(s) requested some additional statistical analyses, which we were forced to describe in our statistical analyses section. In our mind, it is important to describe all the statistical methods needed/used detailed enough. However, we have tried to clarify and condense our statistical analyses and their descriptions.
Note also, that as an e-Journal only, the Animals does neither has any length restrictions related the length of the ms nor the number of references (in fact, there should be at least 40 references in the article type papers in the Animals). However, we have now done our best to condense our writing style, the ms is now about 50 lines shorter than the original version.
We have used Wiley's professional English editing services to correct our English usage, and we have done the minor corrections suggested by the reviewers.

Reviewer 5 Report (New Reviewer)
Comments and Suggestions for Authors
I read the manuscript entitled “Soil Characteristics and Distance to the Water Influencing Nest Burrowing Potential of the Sand Martins”.
Overall, the manuscript has the potential to provide insights into the factors that affect the nesting site preferences of the sand martins. The ms is well written and has the potential to provide valuable information. The data were analyzed properly and the presentation of the data was properly arranged. Below are some comments:
The introduction section should be more focused on the information concerning the sand martin and the information of the study species should be placed in the introduction section rather than the methodology.
Also, the supplementary material should be organized in tables according to the template of the journal and not just the tables from the spss analyses.
Line 59: please write “riparia)”
Lines 75-76: please give information on the population status, conservation status globally and on Turkey, and the main threats that lead to this population decline.
Lines 79-81: write the common names of the species with the first letter capital as you did to other species in the ms
Line 297: the statistical analysis section is in two parts. Please rewrite in one
Line 312: write “the significance.”
Line 335: make comparisons of the general characteristics between the colonies.
In figures 6-9 be more informative on what the figures present (give statistics)
Author Response
Reviewer 5
We thank the reviewer for the valuable comments for our manuscript. We have practically taken care all of them, but we wanted to notice than we have received comments from exceptionally many (five!) reviewers for our ms. Therefore, we have been forced to consider, at some level, the remarks of the all five reviewers at the same time. Our responses for your remarks are given in bold/italics after your comments.
I read the manuscript entitled “Soil Characteristics and Distance to the Water Influencing Nest Burrowing Potential of the Sand Martins”.
Overall, the manuscript has the potential to provide insights into the factors that affect the nesting site preferences of the sand martins. The ms is well written and has the potential to provide valuable information. The data were analyzed properly and the presentation of the data was properly arranged. Below are some comments:
Thank you for your positive views about our manuscript.
The introduction section should be more focused on the information concerning the sand martin and the information of the study species should be placed in the introduction section rather than the methodology.
We have condensed the Introduction section a little bit, and moved some parts for the Discussion section (like parts related to the nest holes, predators, and parasites). We also agreed, that it is better to describe the backgrounds related to the study species in the Introduction, therefore, we have deleted the separate chapter 2.2. Study Species.
Also, the supplementary material should be organized in tables according to the template of the journal and not just the tables from the spss analyses.
We have re-organized the supplementary material as requested; we removed all unnecessary parts away, but save all necessary information related to the tests.
Line 59: please write “riparia)”
It is written as “riparia”.
Lines 75-76: please give information on the population status, conservation status globally and on Turkey, and the main threats that lead to this population decline.
We added the following text: “Breeding sites of the Sand Martin are generally ephemeral, and can often threatened by human activities, like flood control and erosion control on rivers [50].”
Lines 79-81: write the common names of the species with the first letter capital as you did to other species in the ms
Done.
Line 297: the statistical analysis section is in two parts. Please rewrite in one.
Sorry about that, we have removed the duplicated sub-title 2.2. Statistical Analyses. However, the methods itself were not described two times, but we clarified the methods a little bit.
Line 312: write “the significance.”
Corrected.
Line 335: make comparisons of the general characteristics between the colonies.
Unfortunately, it is not possible compare statistically the general structures of the colonies because each colony has e.g. only one height or length of the embankment.
In figures 6-9 be more informative on what the figures present (give statistics)
Results of the statistical tests are given in the Table 2. Figure 5 was removed because of the request of the other reviewer.
We have used Wiley's professional English editing services to correct our English usage, and we have done the minor corrections suggested by the reviewers.

Round 2
Reviewer 4 Report (New Reviewer)
Comments and Suggestions for Authors
The abstract needs a more logical flow.
Introduction should be clearly structured
Habitat selection in general
Use of burrows
Influence of soil properties
The sand martin
Research questions should be shortened and expressed more clearly
Again, in my opinion the two non-lake shore colonies are only one because of their close proximity
Study area – position, climate, description of the colony
The rest of material and methods is clearly structured and understandable
Results
Fig..3 add % to the numbers than the figure is totally clear
The rest of the results are clearly understandable
Discussion
Is it necessary to discuss occupancy rate of the nests – this is not a focus of this paper
The rest of the other factors are discussed quite intensively and in my opinion, this can be shortened to select to most important aspects.
Conclusion
I still think that the conclusion about lake shore and non-lake shore habitats is not supported by these data. The rest of the conclusion is clear to me.
Author Response
Responses for the Reviewer R4 2nd round comments
We thank the reviewer for her/his comments for our manuscript. We have tried to follow all the suggestions given, but at the same time we must note that we must satisfy the views of the other four reviewers. Therefore, we have not edited anymore the structure of the abstract, introduction, and study area description because the other four reviewers have been satisfied for them. In our mind, changing these at this final phase would be unfair for the other four reviewers. We hope that you will understand our view. However, we have done some minor edits that we have highlighted in the yellow overlaying in the manuscript file, and below we respond your remarks by bolding. We think now that we have successfully revised the ms, and hope that the ms will be now publishable for the Animals (MDPI).
The abstract needs a more logical flow.
The abstract has already a logical flow: background, why it is important to study the topic, what we exactly studied, study hypotheses and predictions, results and main findings and management suggestions.
Introduction should be clearly structured
Habitat selection in general
Use of burrows
Influence of soil properties
The sand martin
Research questions should be shortened and expressed more clearly
Again, in my opinion the two non-lake shore colonies are only one because of their close proximity
During the 1st review round we have condensed the Introduction section based on the suggestions of all five reviewers. In fact, the current structure of the Introduction section is basically the same as described over: 1) Nest site selection in general; 2) Factors influencing nesting success of burrowing bird species; 3) Description of the soil factors studies in this research; 4) Description of the Sand Martin; 5) Main study hypotheses and corresponding predictions.
Related to the research questions, other reviewers have asked to described them a more detailed. As stated in the ms, our main study aim was “We studied Sand Martin nesting colonies in the Lake Van area in Turkey and related the morphological characteristics of the nesting burrows to soil characteristics (particle size, pH, and EC).”; We now deleted the “distance to water away from this sentence”. More detailed study hypotheses and predictions are given in L128-142.
Related to the remark that “the two non-lake shore colonies are only one because of their close proximity”; we agree that the two non-lake shore colonies are located quite near of each other’s. However, as you can see from the Table 1, the height and length of the embankments of the non-shore colonies (CV-1 and CV-2) differs, as do the number of holes in these embankments. And as you can see from the Figure 3, also the soil particle size distributions differ between these two colonies. Therefore, according to the our views (note that the other four reviewers have not raised no doubts about this topic), we can considered these two colonies as separate for our study purposes (to study how soil characteristics influence on nest morphology).
Study area – position, climate, description of the colony
Good remark, we have re-ordered this part.
The rest of material and methods is clearly structured and understandable
Thanks
Results
Fig..3 add % to the numbers than the figure is totally clear
Good note, we have now added “%” after the numbers, i.e. the figure is now clearer.
The rest of the results are clearly understandable
Thanks
Discussion
Is it necessary to discuss occupancy rate of the nests – this is not a focus of this paper
We agree that to study occupancy rate of the nests was not in the main focus of our paper. However, we think that it is good to report them because there are only a few earlier studies that have published such information. Also, occupancy rate will indicate the suitability of the embankment and its´s soil structure for the Sand Martins. Therefore, we will keep this short background part in our paper.
The rest of the other factors are discussed quite intensively and in my opinion, this can be shortened to select to most important aspects.
The detailed discussion is partly related to the questions and suggestions raised by the other four reviewers. Note also, that because of the suggestions of the other reviewers, we replaced some material from the Introduction section to the Discussion section. However, because of your request, we have now condensed the Discussion section a little bit, and we have deleted the following material away:
“It is important to note that the size of these particles was largely influenced by local soil composition and geological factors.
It is possible that this condition affected the soil structure. Interestingly, all the soil samples we collected from the Sand Martin colonies had particle sizes of 0.106 mm (with a proportion of over 70% in all colonies).
The areas where Sand Martins form nest colonies around the shore of Lake Van have similar habitat characteristics because the birds typically prefer to breed in sandy or muddy banks or cliffs. The fact that the soil content of the nesting holes in the different colonies consists mostly of fine dunes is consistent with this preference, because dunes are often composed of fine, loose sand that is relatively easy to excavate.
According to the soil particle analysis results for the colonies, the soil samples obtained from CV-1 to DMS were sandy, and most of the particles consisted of 0.425 mm (79%) granular structures, while the CV-2 and DMS colonies’ samples were sandy, and the majority of the particles consisted of 0.106 m (81%) particulate structures.
Our results indicated a significant difference in the width of the entrance opening of the nesting tunnels of Sand Martins between two of the colonies studied (CV2–DMS and CV2–DMC. The results showed that the average values of the lakeshore colonies (DMS, DMC) concerning the width of the tunnel entrance opening were higher than those of the other colonies (CV1, CV2). The differences in the width of the nest tunnel entrance opening between colonies may be due to the location of the colony. Among the colonies examined, CV-1 and CV-2 were located far from the lake shore, whereas DMS and DMC colonies were located very close to the lake shore.”
In addition, we split some long paragraphs to increase the readability of the discussion section.
Conclusion
I still think that the conclusion about lake shore and non-lake shore habitats is not supported by these data. The rest of the conclusion is clear to me.
As your request, we now changed the following sentence “Our results indicated that the location of the colony (lake-shore site vs. non-lake shore site) was a critical predictor of the physical properties of the nesting holes, including tunnel depth and space between tunnel holes.”
as
“Our results indicated that the location of the colony was a critical predictor of the physical properties of the nesting holes, including tunnel depth and space between tunnel holes.”
On the behalf of the all co-authors
EC

This manuscript is a resubmission of an earlier submission. The following is a list of the peer review reports and author responses from that submission.
Round 1
Reviewer 1 Report
Comments and Suggestions for Authors
It is an interesting and valuable paper. Materials and methods are presented properly. Finding are interesting however, I have some comments that are presented in the pdf version of the manuscript. My major suggestion is to rearrange some parts of the paper: discussion is too long- authors should give more clear and straighforward explanations for their findings, some information from the results section should be delated (eg. list of mean values that are presented in the tables), other could be presented using figures. Moreover, authors did not analyse preferences as they did not have information/data from locations that were not occupied by birds. If you talk about preferences you should compare locations with birds and without birds.

Language is rather proper and sentences area easy to follow.
Reviewer 2 Report
Comments and Suggestions for Authors
The topics is interest but based on a small sample. Inaccurately described ways of random selection for measurements, statistical methods need to be improved. Discussion is very long, many threads raised were not studied, needs to be significantly shortened. Too long list of literature, citation of some papers is not justified.

Quality of English should be improved.
Reviewer 3 Report
Comments and Suggestions for Authors
